# EFFICIENT GRADIENT ESTIMATION VIA ADAPTIVE SAMPLING AND IMPORTANCE SAMPLING

## ABSTRACT

Machine learning problems rely heavily on stochastic gradient descent (SGD) for optimization. The effectiveness of SGD is contingent upon accurately estimating gradients from a mini-batch of data samples. Instead of the commonly used uniform sampling, adaptive or importance sampling reduces noise in gradient estimation by forming mini-batches that prioritize crucial data points. Previous research has suggested that data points should be selected with probabilities proportional to their gradient norm. Nevertheless, existing algorithms have struggled to efficiently integrate importance sampling into machine learning frameworks. In this work, we make two contributions. First, we present an algorithm that can incorporate existing importance functions into our framework. Second, we propose a simplified importance function that relies solely on the loss gradient of the output layer. By leveraging our proposed gradient estimation techniques, we observe improved convergence in classification and regression tasks with minimal computational overhead. We validate the effectiveness of our adaptive and importance-sampling approach on image and point-cloud datasets.

## 1 INTRODUCTION

Stochastic gradient descent (SGD) combined with back-propagation and efficient gradient techniques—such as Adam [12]—has unlocked a realm of possibilities. Its importance lies in its ability to optimize complex models by iteratively updating parameters based on the gradient of the loss function. However, despite its significance, SGD has notable limitations. Convergence rates depend on various factors, notably the noise in gradient estimation, which significantly impacts both robustness and speed. Addressing this noise, and effectively reducing it, is an active area of research [1; 5; 9; 6; 16].

Several approaches have been proposed to mitigate the noise in gradient estimation, including data diversification [26; 27], adaptive batch sizes, weighted sampling [19] or importance sampling [10]. These methods aim to improve the quality of gradient approximations and accelerate convergence in noisy optimization landscapes.

Our proposed idea centers around the concept of importance sampling, which involves constructing mini-batches using a non-uniform data-point selection, i.e., selecting certain data points with higher likelihood. The objective is to strategically allocate computational resources to data points that exert the most significant influence on the optimization task.

In this paper, we introduce a novel approach that leverages information extracted from the network's output to quantify the extent of model modification required for each data sample. This information guides our importance sampling strategy, resulting in substantial improvements in convergence across various tasks. Our approach has a lower computational overhead compared to state-of-the-art methods [10; 19].

In summary, our contributions can be distilled into the following key points:

- We propose an efficient and robust strategy for adaptive and importance sampling.

- We introduce an algorithm with minimal overhead for implementing adaptive and importance sampling.

- We demonstrate the effectiveness of our approach across classification and regression problems, encompassing both importance sampling and adaptive sampling techniques.

## 2 RELATED WORK

Gradient estimation serves as a cornerstone in the realm of machine learning, underpinning the optimization of models. In practical scenarios, computing the exact gradient is often unfeasible due to the sheer volume of data, leading to the reliance on mini-batch approximations. Improving these approximations to obtain more accurate and lower-variance estimates is an enduring challenge in the field. The ultimate goal is to expedite gradient descent optimization by enhancing the quality of gradient approximations, representing a core objective in the domain of machine learning.

**Importance sampling.** Sampling data points proportionally to the norm of the gradient is regarded as the optimal choice in optimization. Bordes et al. [2] developed an online algorithm (LASVM) that uses importance sampling to train kernelized support vector machines. Zhao & Zhang [29]; Needell et al. [16]; Wang et al. [24]; Alain et al. [1] proved that importance sampling proportional to the gradient norm is the optimal sampling strategy. However, even with this optimal strategy, the resulting estimation is not error-free as multiple parameters derivatives are estimated simultaneously. Nevertheless, this approach represents a practical trade-off that provides a good compromise in terms of optimization quality across all dimensions simultaneously.

In practice, estimating the gradient for each data point can be computationally intensive. This has prompted the search for more efficient sampling strategies that can approximate the gradient norm without incurring excessive computational costs. Ideally, such alternative strategies should mimic the gradient norm's behavior while remaining computationally lightweight, thus contributing to the scalability and practicality of machine learning algorithms Katharopoulos & Fleuret [10]. Loshchilov & Hutter [14] proposed to use a ranking data based on their respective loss. This rank is then use to create an importance sampling strategy giving more importance to data with high rank (high loss). Katharopoulos & Fleuret [11] proposed to importance sample the loss function. Dong et al. [3] proposed a re-sampling based algorithm to reduce the number of backpropagation computation. A sub-selection of data is performed based on the loss. Similarly, Zhang et al. [28] proposed to use a re-sampling based on multiple heuristics to reduce the number of backward propagations and focus on more contributing data. Katharopoulos & Fleuret [10] proposed an upper bound to the gradient norm that can be used as an importance function. They proposed to re-sample data based on importance computed on the last layer. All the re-sampling method reduce the number of unnecessary backward propagation but still suffer from forward computation.

We propose an efficient algorithm and an importance function which when used for importance or adaptive sampling, shows significant improvements.

**Adaptive data weighting/sampling.** Adaptive weighting/sampling operates by dynamically adjusting the contribution of individual data samples during optimization. This dynamic alteration of weights aims to prioritize specific data points that exert more influence on the gradient descent process. This alteration can be performed by either increasing or decreasing their weight contributions to the estimator. Adaptive sampling is different from importance sampling, primarily in the way the gradient estimator is defined for each respective strategy. While adaptive weighting/sampling significantly accelerates convergence, it does so by introducing a bias into the gradient estimation.

To compute adaptive weights within a given mini-batch, Santiago et al. [19] proposed a method that maximizes the effective gradient of the mini-batch. This approach strategically allocates weights to data points, aligning their contributions with the optimization objective. While this may introduce bias, it allows for a faster and more efficient convergence towards an optimal solution, making adaptive weighting a valuable strategy in machine learning optimization. Our proposed algorithm and the corresponding adaptive weighting approach naturally focuses on data points with high sampling probabilities.

## 3  Gradient estimation

In machine learning, optimization is key to refining the models. The goal is to find optimal parameters $\theta$ for a model function $m(x, \theta)$, with $x$ a data sample, that minimize a loss function $\mathcal{L}$ over a dataset $\Omega$. The optimization is typically expressed as

$$\theta^* = \underset{\theta}{\arg\min}\, L_\theta, \quad \text{where} \quad L_\theta = \frac{1}{|\Omega|} \int_\Omega \mathcal{L}(m(x, \theta), y)\,\mathrm{d}x. \tag{1}$$

The loss function $\mathcal{L}$ quantifies the difference between model predictions $m(x, \theta)$ and actual data $y$. The factor in front of the integral normalizes the total loss $L_\theta$ by the dataset size.

In practice, the minimization of the total loss is tackled via iterative gradient descent. At each step $t$, the gradient $\nabla L_{\theta_t}$ of that loss with respect to the current model parameters $\theta_t$ is computed, and the parameters are updated as

$$\theta_{t+1} = \theta_t - \lambda \nabla L_{\theta_t}, \tag{2}$$

where $\lambda > 0$ is the learning rate. This iterative procedure can be repeated until convergence.

### 3.1  Monte Carlo gradient estimator

The parameter update step in Eq. (2) involves evaluating the total-loss gradient $\nabla L_{\theta_t}$. This requires processing the entire dataset $\Omega$ at each of potentially many (thousands of) steps, rendering the optimization computationally infeasible. In practice one has to resort to mini-batch gradient descent which estimates the gradient from a small set $\{x_i\}_{i=1}^B \subset \Omega$ of randomly chosen data points in a Monte Carlo fashion:

$$\nabla L_\theta \approx \frac{1}{|\Omega| \cdot B} \sum_{i=1}^B w(x_i) \nabla \mathcal{L}(m(x_i, \theta), y_i) = \langle \nabla L_\theta \rangle, \quad \text{with} \quad x_i \propto p(x_i). \tag{3}$$

Here, $\nabla \mathcal{L}(m(x_i, \theta), y_i)$ is the gradient (w.r.t. $\theta$) of the loss function for sample $x_i$ selected following a probability density function (pdf) $p$ (or probability mass function in case of a discrete dataset). Setting the weighting function to $w(x_i) = 1/p(x_i)$ makes $\langle \nabla L_\theta \rangle$ an unbiased estimator for the total loss, i.e., $\mathrm{E}[\langle \nabla L_\theta \rangle] = \nabla L_\theta$. Mini-batch gradient descent uses $\langle \nabla L_\theta \rangle$ in place of the true gradient $\nabla L_\theta$ in Eq. (2) to update the model parameters at every iteration. The batch size $B$ is typically much smaller than the dataset, enabling practical optimization. To preserve energy, we impose the condition $E[w(x)] = |\Omega|$, signifying that the average weight value should equal the cardinality of $\Omega$, which represents the number of elements in the dataset. It ensure the weight $w(x)$ compensate for the factor $1/|\Omega|$ in front of the summation.

### 3.2  Sampling and weighting strategies

Mini-batch gradient notoriously suffers from Monte Carlo noise in the gradient estimate (3), which can make the parameter-optimization trajectory erratic and convergence slow. That noise comes from the often vastly different contributions of different samples $x_i$ to that estimate. Oblivious to this, classical mini-batch gradient descent assigns equal importance to all samples, selecting and weighting them uniformly. Several strategies can be employed to efficiently and effectively reduce the estimation noise, based on judicious, non-uniform sample selection/weighting.

**Importance sampling**  Typically, the selection of samples that go into a mini-batch is done with constant probability $p(x_i) = 1/|\Omega|$. Importance sampling is a technique for using a non-uniform pdf to strategically pick samples proportionally on their contribution to the gradient, to reduce estimator variance. Setting $w(x_i) = 1/p(x_i)$ maintains unbiasedness for any valid importance distribution $p$. A distribution $p$ is valid if it assigns non-zero probability to samples with non-zero contribution.

**Adaptive sampling.**  Adaptive sampling represents an alternative approach to data selection, where samples are chosen based on non-uniform sampling distributions without applying normalization to their contributions. In this method, as outlined in Eq. (3), each data point's weight, denoted as $w(x_i)$, remains constant at $|\Omega|$. Consequently, adaptive sampling naturally focuses on data points with high sampling probabilities, effectively modifying the problem statement Eq. (1) by defining a revised

domain $\Omega'$. This domain is denser around data points with elevated sampling probabilities, making it adaptable to evolving probabilities during the optimization process.

Adaptive weighting employs a uniform sampling distribution, where $p(x_i) = 1/|\Omega|$, but introduces an adaptive weight normalization factor, $w(x_i)$. In contrast to adaptive sampling, which focuses on non-uniform data selection, adaptive weighting prioritizes data samples by assigning them varying levels of importance through the weight normalization. Much like its counterpart, this method results in a modification of the problem statement, allowing for the emphasis of specific data samples during the optimization process. If carefully chosen, this emphasis can significantly accelerate optimization convergence.

### 3.3 PRACTICAL IMPORTANCE/ADAPTIVE SAMPLING ALGORITHM

We propose an algorithm to efficiently perform importance and adaptive sampling for mini-batch gradient descent, outlined in Algorithm 1. Similarly to Loshchilov & Hutter [14] and Schaul et al. [20], it is designed to use an importance function that relies on readily available quantities for each data point, introducing only negligible memory and computational overhead over classical uniform mini-batching.

---

**Algorithm 1** Gradient descent using importance or adaptive sampling

---

1:   $\theta \leftarrow$ random parameter initialization
2:   $B \leftarrow$ mini-batch size, $|\Omega| \leftarrow$ dataset size
3:   $\alpha \leftarrow$ importance momentum coefficient
4:   $q \leftarrow \mathbf{1}$              $\leftarrow$ persistent per-sample importance
5:
6:   $q, \theta \leftarrow \textsc{Initialisation}(x, y, \Omega, \theta, B, q)$        $\leftarrow$ Algorithm 3 in Appendix B
7:
8:   **until** convergence **do**               $\leftarrow$ loop over epochs
9:      **for** $t \leftarrow 1$ **to** $|\Omega|/B$ **do**        $\leftarrow$ loop over mini-batches
10:        $p \leftarrow q/\text{sum}(q)$            $\leftarrow$ compute the pdf from q
11:        $x, y \leftarrow B$ data samples $\{x_i, y_i\}_{i=1}^{B}$ chosen proportionally to $p$
12:        $l(x) \leftarrow \mathcal{L}(m(x, \theta), y)$
13:        $\nabla l(x) \leftarrow \text{Backpropagate}(l(x))$
14:        $w(x) \leftarrow (\text{ImportanceSampling}) \, ? \, 1/p(x) : |\Omega|$    $\leftarrow$ weight for adaptive or importance sampling
15:        $\langle \nabla L_\theta \rangle (x) \leftarrow (\nabla l(x) \cdot w(x)^T)/(B \cdot |\Omega|)$       $\leftarrow$ Eq. (3)
16:        $\theta \leftarrow \theta - \eta \langle \nabla L_\theta \rangle (x)$           $\leftarrow$ Eq. (2)
17:        $q(x) \leftarrow \alpha \cdot q(x) + (1-\alpha) \cdot \textsc{ComputeSampleImportance}(x, y)$ $\leftarrow$ Eq. (7) and Algorithm 2
18:     $q \leftarrow q + \epsilon$        $\leftarrow$ small positive $\epsilon$ added to ensures no data samples are forgotten indefinitely
19:
20: **return** $\theta$

---

We store a set of persistent *un-normalized importance* scalars $q = \{q_i\}_{i=1}^{|\Omega|}$ that are updated continuously during the optimization. We begin with a first epoch to processes all data points once and determine their initial importance (line 6). After that, at each mini-batch optimization step $t$, we normalize the importance values to obtain the probability density function (PDF) $p$ (line 10). We then extract $B$ data samples (with replacement) using this PDF $p$ (line 11). The loss $\mathcal{L}$ is evaluated for each selected data sample (line 12), and backpropagated to compute the corresponding loss gradient (line 13). Depending on whether we want to perform importance sampling or adaptive sampling, the per-sample weight is selected (line 14) and used in the gradient estimation (3) (line 15). For importance sampling $w(x) = 1/p(x)$ and for adaptive sampling $w(x) = |\Omega|$. Finally, the network parameters are updated using the estimated gradient (line 16). In line 17, the subroutine $\textsc{ComputeSampleImportance}(x, y)$ returns the sample importance for each data sample from the mini-batch. Any importance heuristic such as the gradient norm [29; 16; 24; 1] or the loss [14; 11; 3], or more advanced importance [10] can be implemented in this subroutine. For efficiency, our algorithm reuses the forward pass computations made during line 12 to execute $\textsc{ComputeSampleImportance}(x, y)$ subroutine, thereby updating $q$ only for the current mini-batch samples. The weighting parameter $\alpha$ ensures weight stability as discussed in Eq. (7). At the end of each epoch (line 18), we add a small value to the un-normalized weights of all data to en-

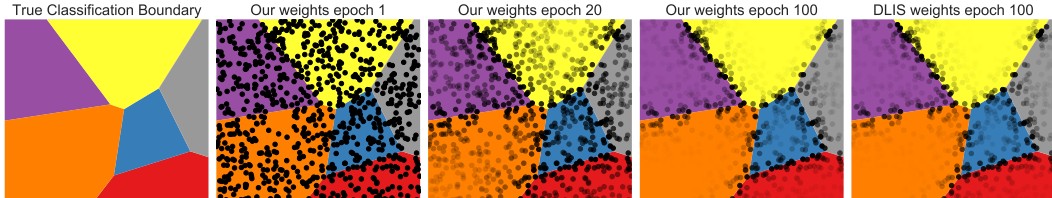

Figure 1: Visualization of the importance sampling at 3 different epoch and the underlying classification task. For each presented epoch, 800 data-point are presented with a transparency proportional to their weight according to our method.

sure that every data point will be eventually evaluated, even if its importance is deemed low by the importance metric.

It is importance to note that the first epoch is done without importance sampling to initialize each sample importance. This does not create overhead as it is equivalent to a classical epoch running over all data samples. While similar schemes have been proposed in the past, they often rely on a multitude of hyperparameters, making their practical implementation challenging. This has led to the development of alternative methods like re-sampling [10; 3; 28]. Our proposed sampling strategy has only a few hyperparameters. Tracking importance across batches and epochs minimizes the computational overhead, further enhancing the efficiency and practicality of the approach.

## 4 LOSS-GRADIENT-BASED IMPORTANCE FUNCTION

---
**Algorithm 2** subroutine for cross entropy loss importance metric

---
1: **function** COMPUTESAMPLEIMPORTANCE($x_i$,$y_i$)     ← $x_i$ = data sample, $y_i$ = class index of $x_i$
2:     $s = \exp(m(x_i, \theta)) / \sum_{k=0}^{J} \exp(m(x_i, \theta)_k)$     ← Eq. (4)
3:     $q = \sum_{j=1}^{J} s_j - \mathbf{1}_{j=y_i}$     ← Eq. (5)
4:     **return** $q$

---

Classification assigns discrete labels to data points, relying on identifying complex decision boundaries in the feature space. Accurate boundary identification is crucial, as minor parameter changes can significantly impact results. Importance sampling in classification emphasizes gradients along these boundaries, where parameter modifications have the greatest impact. Figure 1 illustrates this concept, showing iterative refinement of the sampling distribution to focus on boundary decisions in comparison to data within classes. The rightmost column illustrates the sampling distribution of the DLIS method of Katharopoulos & Fleuret [10] at epoch 100. Both methods iteratively increase the importance of the sampling around the boundary decision compare to data inside the classes.

Our approach differs from that of Katharopoulos & Fleuret in that we compute the gradient norm with respect to the network's output logits. This approach often allows gradient computation without requiring back-propagation or graph computations, streamlining optimization.

**Cross-entropy loss gradient.** Cross entropy is a widely used loss function in classification tasks. It quantifies the dissimilarity between predicted probability distributions and actual class labels. Specifically, for a binary classification task, cross entropy is defined as:

$$\mathcal{L}(m(x_i, \theta)) = -\sum_{j=1}^{J} y_j \log(s_j) \text{ where } s_j = \frac{\exp(m(x_i, \theta)_j)}{\sum_{k=0}^{J} \exp(m(x_i, \theta)_k)} \tag{4}$$

where $m(x_i, \theta)$ is an output layer, $x_i$ is the input data and $J$ means the number of classes. The derivative of the loss $\mathcal{L}$ of a data point $x_i$ with respect to the network output layer $m(x_i, \theta)_j$ reads

$$\frac{\partial \mathcal{L}}{\partial m(x_i, \theta)_j} = s_j - y_j \tag{5}$$

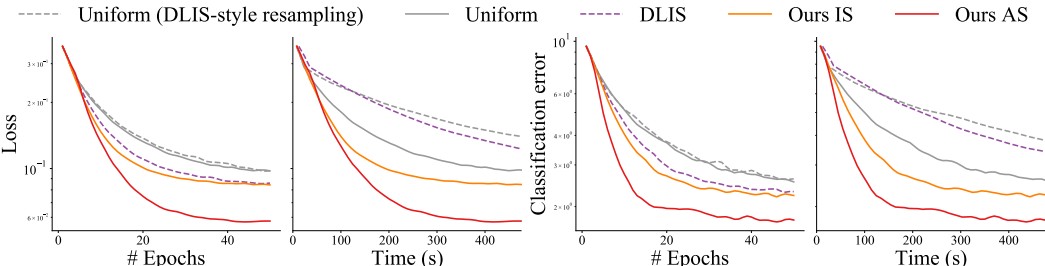

Figure 2: We compare loss and classification error metrics for the MNIST dataset between the resampling algorithm by Katharopoulos & Fleuret [10] (DLIS) and our algorithm. At equal epochs, the resampling algorithm with importance sampling works better than uniform sampling for DLIS. However, at equal time, the resampling cost is too high, making DLIS even slower than standard uniform sampling. Our algorithm outperforms all existing methods.

This equation can be directly computed from the network output without any graph back-propagation. Algorithm 2 show a short pseudo-code of the importance computation for Algorithm 1. Proof of the derivation can be found in the Appendix A.

Similarly to the approach of Katharopoulos & Fleuret [10], the norm of the gradient across the network can be bounded as follows:

$$\left\| \frac{\partial \mathcal{L}(x_i)}{\partial \theta} \right\| = \left\| \frac{\partial \mathcal{L}(x_i)}{\partial m(x_i, \theta)} \cdot \frac{\partial m(x_i, \theta)}{\partial \theta} \right\| \leq \left\| \frac{\partial \mathcal{L}(x_i)}{\partial m(x_i, \theta)} \right\| \cdot \left\| \frac{\partial m(x_i, \theta)}{\partial \theta} \right\| \tag{6}$$

While this bound may not offer conclusive proof of optimality or a direct correlation with the gradient norm, it establishes a discernible link between the two. The key lies in the initial gradient's selection at the start of the chain rule. This gradient essentially sets the tone for all subsequent gradients in the network. Consequently, when variations in this initial gradient of the loss are pronounced, energy propagates through the network, elevating the gradient norm. Essentially, the intricacies of gradient variation at the outset have a rippling effect, significantly influencing the overall gradient norm, despite the bound not providing a definitive measure of optimality.

**Generalized loss gradient.** We can extend our framework to other problems, e.g., regression, using the automatic gradient computation operator. Our framework can compute the loss gradient w.r.t. the output layer using such autograd operators. To demonstrate the generalization capacity of our framework, we also perform experiments on the regression problem.

## 5 EXPERIMENTS

In this section, we delve into the experimental outcomes of our proposed algorithm and sampling strategy. Our evaluations encompass diverse classification tasks, spanning both importance sampling and adaptive sampling. We benchmarked our approach against those of Katharopoulos & Fleuret [10] and Santiago et al. [19], considering various variations in comparison. Distinctions in our comparisons lie in assessing performance at equal steps/epochs and equal time intervals. The results presented here demonstrate the loss and classification error, computed on test data that remained unseen during the training process.

### 5.1 IMPLEMENTATION DETAILS

For fair comparisons, we implement our method and all baselines in a single PyTorch framework. Experiments run on a workstation with an NVIDIA GeForce RTX 2080 graphics card and an Intel(R) Core(TM) i7-9700 CPU @ 3.00GHz. The baselines include uniform sampling, DLIS [10] and LOW [19]. Uniform means we sample every data point from a uniform distribution. DLIS importance samples the data mainly depending on the norm of the gradient on the last output layer. We use functorch [8] to accelerate this gradient computation. LOW is based on adaptive weighting that maximizes the effective gradient of the mini-batch using the solver in Vandenberghe [23].

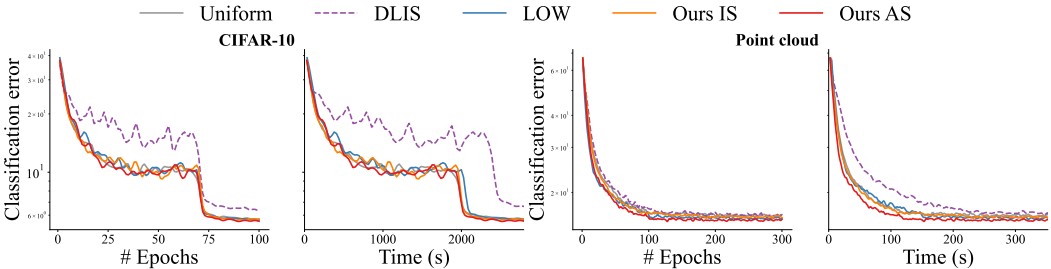

Figure 3: When comparing on CIFAR-10 and Point cloud ModelNet40 [25] classification datasets, DLIS performs poorly at equal time due to the resampling overhead. Unlike DLIS, we use standard uniform sampling which is faster. We also compare against another adaptive scheme by Santiago et al. [19] (LOW). Our adaptive (Ours AS) and importance sampling (Ours IS) shows improvements on the ModelNet40 dataset against other methods. Overall, our adaptive variant achieves lower classification errors with minimal overhead compared to others.

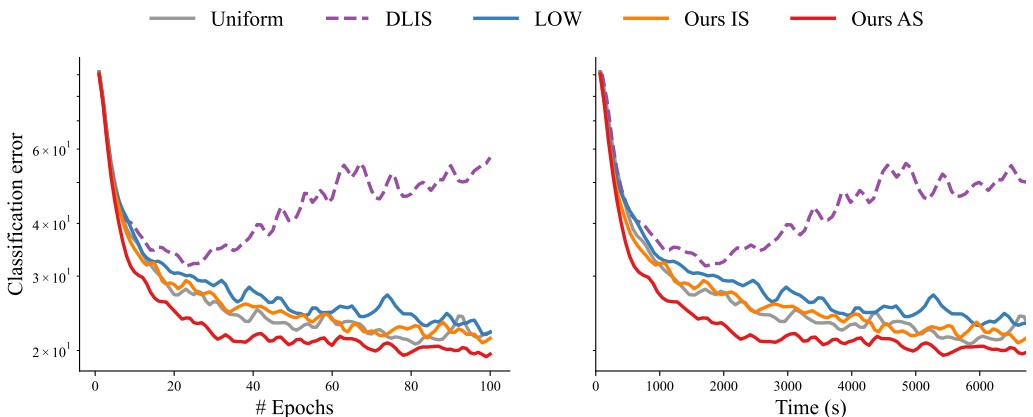

Figure 4: On Oxford flower 102 classification dataset [17], where the number of classes are high, our approach shows significant improvement compared to other methods. DLIS performs worse due to the sparsity in the data, which hampers their resampling strategy.

**Weight stability.** Updating the persistent per-sample importance $q$ directly sometime leads to a sudden decrease of accuracy during training. To make the training process more stable, we update $q$ by linearly interpolating the importance at the previous and current steps:

$$q(x) = \alpha \cdot q_{prev}(x) + (1 - \alpha) \cdot q(x) \tag{7}$$

where $\alpha$ is a constant for all data samples. In practice, we use $\alpha \in \{0.0, 0.1, 0.2, 0.3\}$ as it gives the best trade-off between importance update and stability. This can be seen as a momentum evolution of the per-sample importance to avoid high variation. Utilizing an exponential moving average to update the importance metric prevents the incorporation of outlier values. This is particularly beneficial in noisy setups, like situations with a high number of class or a low total number of data.

**MNIST dataset.** The MNIST database contains 60,000 training images and 10,000 testing images. We train a 3-layer fully-connected network (MLP) for image classification over 50 epochs with an Adam optimizer [12].

**CIFAR dataset.** CIFAR-10 Krizhevsky et al. [13] contains 60,000 32x32 color images from 10 different object classes, with 6,000 images per class. CIFAR-100 Krizhevsky et al. [13] has 100 classes containing 600 images each, with 500 training images and 100 testing images per class. For both datasets, we train the same ResNet-18 network [7]. We use the SGD optimizer with momentum 0.9, initial leaning rate 0.01, and batch size 64. We divide the initial learning rate by 10 after 70

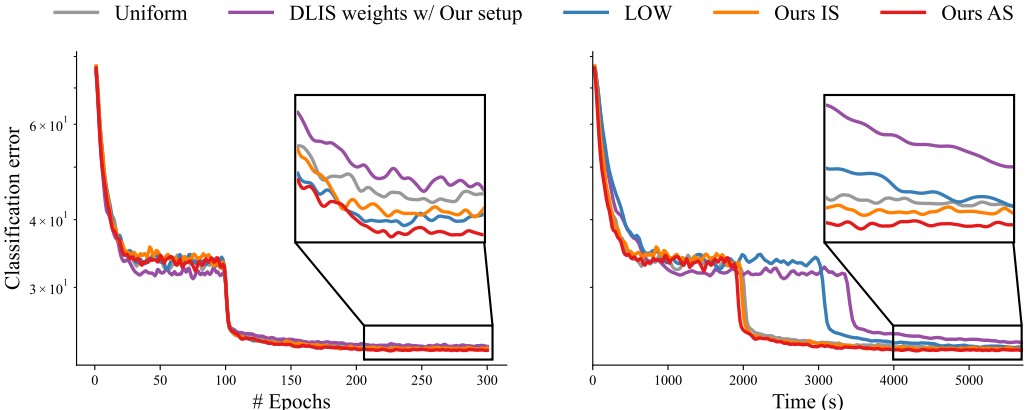

Figure 5: On CIFAR-100 classification dataset, instead of comparing the DLIS resampling algorithm, we use DLIS weights in our algorithm. We display zoom-in of the end of the curves to highlight the differences. At equal epochs (left), our methods (Ours IS & AS) show improvements compared to LOW [19] and DLIS weights. At equal time (right), LOW and the DLIS weights takes longer to converge. Overall our approach shows faster convergence with lower importance computation.

epochs for CIFAR-10 and train the network for a total of 100 epochs. Additionally, we trained a Vision Transformer (ViT) [4] on CIFAR-10 using the Adam optimizer with an initial learning rate 0.0001 and a cosine annealing scheduler [15]. For CIFAR-100, we divide the learning rate by 10 after 100, 200 epochs and train for a total of 300 epochs. For both datasets, we use random horizontal flip and random crops to augment the data on the fly.

**Point-cloud classification dataset.** We train a PointNet Qi et al. [18] with 3 shared-MLP layers and one fully-connected layer, on the ModelNet40 dataset Wu et al. [25]. The dataset contains point clouds from 40 categories. The data are split into 9,843 for training and 2,468 for testing. Each point cloud has 1,024 points. We use the Adam optimizer Kingma & Ba [12], with batch size 64, weight decay 0.001, initial learning rate 0.00002 divided by 10 after 100, 200 epochs. We train for 300 epochs in total.

**Oxford 102 flower dataset.** The Oxford 102 flower dataset Nilsback & Zisserman [17] contains flower images from 102 categories. We follow the same experiment setting of Zhang et al. [26; 27]. We use the original test set for training (6,149 images) and the original training set for testing (1,020 images). In terms of network architecture, we use the pre-trained VGG-16 network Simonyan & Zisserman [21] for feature extraction and only train a two-layer fully-connected network from scratch for classification. We use the Adam optimizer Kingma & Ba [12] with a learning rate 0.001 and train the two-layer fully-connected network for 100 epochs.

## 5.2 RESULTS

In Fig. 2, we compare our algorithm against DLIS [10]. DLIS applies resampling to both uniform and their importance sampling. This increases the overall computation overhead for their approach. Standard uniform sampling is much faster than the resampling approach. We assess cross-entropy loss and classification error in terms of epoch count and equal time. The DLIS method shows similar performance to ours at equal epochs but incurs high computational costs due to the need for a large dataset forward pass during re-sampling.

In Fig. 3, we compare uniform sampling, DLIS, LOW, and our importance and adaptive sampling strategies for CIFAR-10 and point cloud classification tasks. Our adaptive sampling excels in classification error across epochs and optimization time, especially when compared at equal time due to our low overhead compared to DLIS and LOW. DLIS performs worse than uniform sampling even with an equal number of epochs for CIFAR-10, highlighting the challenges in effective re-sampling

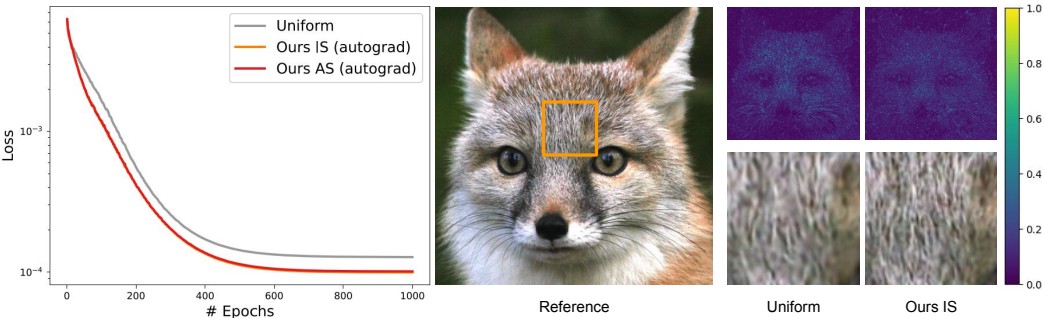

Figure 6: Image regression using a 5-layer SIREN network Sitzmann et al. [22], which is trained for predicting RGB values from 2D pixel coordinates. Our importance and adaptive sampling strategies based on autograd gradients show clear improvements compared to uniform sampling at equal-epoch loss curves. We further show the error map and zoom-in results using uniform sampling and our importance sampling. Equal-time comparisons show similar improvements (see appendix).

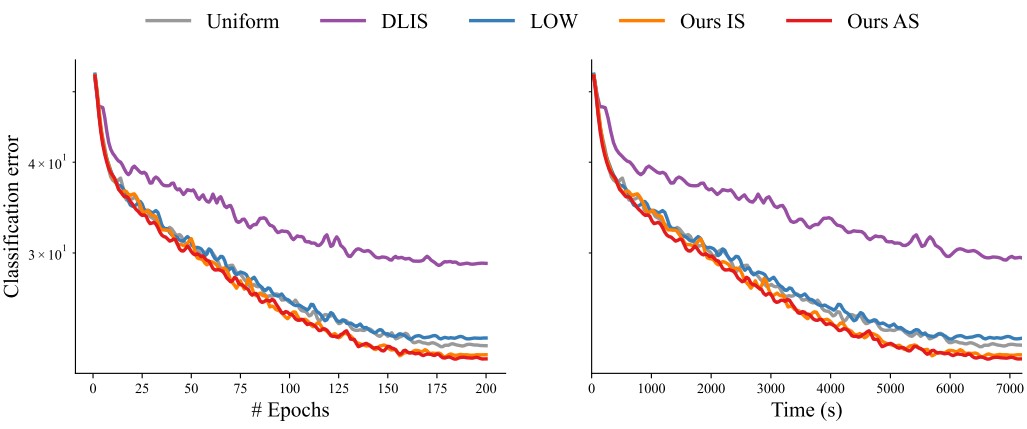

Figure 7: Comparisons on CIFAR-10 using Vision Transformer (ViT) [4]. The results show consistent improvement of Ours IS/AS over LOW [19] and DLIS [10].

strategies, even with 5-fold increase in data sampling. Our algorithm is versatile, can work on different data-formats (images, point clouds). In Fig. 7, we also perform a comparison with a Vision Transformer architecture [4] on CIFAR-10 dataset, where our approach is showing consistent improvements. DLIS convergence is hampered due to resampling which discards many *less important* data samples, causing training to focus on a subset of the dataset.

We conduct a similar experiment on the Oxford flower classification task in Fig. 4). This dataset is known for its complexity due to a large number of classes with few images per class. Our method employing adaptive sampling again emerged as the top performer. Notably, DLIS exhibits under-performance, likely due to the challenges of re-sampling in a dataset with a high number of classes with only (10 data samples per class). With this distribution of data, re-sampling does not achieve a good estimation of the gradient. This causes visible over-fitting in the convergence curve. This highlights the robustness of our sampling metric as well as the use of memory based algorithm.

Our algorithm allows using weights from any existing method as an importance function. To demonstrate this feature, in Fig. 5 we use DLIS weights in our algorithm. Here we present a comparison of classification error for CIFAR-100 dataset. Although all methods perform comparably at equal epochs, Ours AS stands out with the best results. At equal time, both DLIS and LOW are hampered by their respective weight computation overhead, leading to slightly inferior performance. It should be noted that DLIS can be implemented through our algorithm and can benefit from the memory even if there is still an overhead in the weight computation.

Transitioning from re-sampling to a memory-based algorithm proves advantageous in such scenarios. Additional comparisons with the same setup can also be found for MNIST in Appendix C.

Zhao & Zhang [29] have shown that importance weights wrt the gradient norm gives the optimal sampling distribution. On the right inline figure, we show the difference between various weighting strategies and the gradient norm wrt all parameters. In this experiment, all sampling weights are computed using the same network on an MNIST optimization task. Our proposed sampling strategies, based on the loss gradient are the closest approximation to the gradient norm.

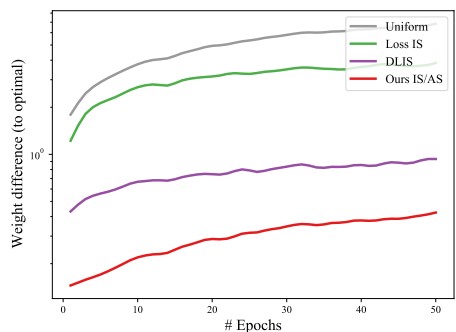

All the results shown have used our analytic importance function from Eq. (5). To show that our importance function can generalize to other problems, we use a simple autograd operation to compute the loss gradient at the output layer. We show it's applicability on the image regression problem in Fig. 6. In this task, we trained a 5-layer SIREN network Sitzmann et al. [22] to predict RGB values from given 2D pixel coordinates. The left side of the figure illustrates the loss evolution wrt the training epochs. On the right, we showcase the reference image we train the network on, the error image (difference from the reference), and a crop for both Uniform sampling and our importance sampling method. To use our method in this example with an $L_2$ norm loss function, we employed autograd computation with respect to the output of the network. Appendix C provides a more detailed comparison between analytic formulation and the autograd for the cross-entropy loss. Our method exhibits better loss convergence despite a minor computational overhead. The error image shows more uniform and reduced amplitude errors compared to uniform sampling. This difference is evident in the crop, where our method provides sharper fur details, contrasting with the blurred result of uniform sampling. This experiment highlights the versatility of non-uniform sampling, expanding its use beyond classification tasks.

**Discussions**   Both ours and DLIS importance metrics are highly correlated but ours is simpler and efficient to evaluate. The efficiency comes from using the memory vector of the importance metric which avoids the need of resampling. The resulting algorithm gives higher performance at equal time and has more stable convergence. Our adaptive sampling achieve better convergence properties than LOW. Adaptive sampling generally outperforms adaptive weighting. Using adaptive sampling mini-batch are composed with multiple high importance data instead of a weighting based on the respective contribution. Thus the average in Eq. (3) is done with more contributing data. Additionally, our loss derivative-based weighting is closer to the gradient norm than loss-based importance.

**Limitations**   As the algorithm rely on past information to drive a non-uniform sampling of data, it requires seeing the same data multiple times. This creates a bottleneck for architectures that rely on progressive data streaming. More research is needed to design importance sampling algorithms for data streaming architectures, which is a promising future direction. Non-uniform data sampling can also create slower runtime execution. The samples selected in a mini-batch are not laid out contiguously in memory leading to a slower loading. We believe a careful implementation can mitigate this issue.

## 6  CONCLUSION

In conclusion, our work introduces an efficient sampling strategy for machine learning optimization, including both importance and adaptive sampling. This strategy, which relies on the gradient of the loss and has minimal computational overhead, was tested across various classification as well as regression tasks with promising results. Our work demonstrates that by paying more attention to samples with critical training information, we can speed up convergence without adding complexity. We hope our findings will encourage further research into simpler and more effective importance/adaptive sampling strategies for machine learning.

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

## A  DERIVATIVE OF CROSS-ENTROPY LOSS

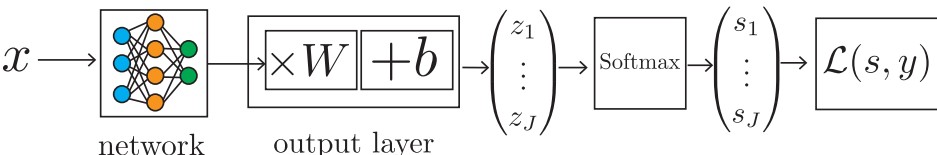

Machine learning frameworks take data $x$ as input, performs matrix multiplication with weights and biases added. The output layer is then fed to the softmax function to obtain values $s$ that are fed

to the loss function. $y$ represents the target values. We focus on the categorical cross-entropy loss function for the classification problem (with $J$ categories) given by:

$$\mathcal{L}_{\text{cross-ent}} = -\sum_i y_i \log s_i \ \text{ where } s_i = \frac{\exp(m(x_i,\theta)_l)}{\sum_l^J \exp(m(x_i,\theta)_l)} \tag{8}$$

For backpropagation, we need to calculate the derivative of the $\log s$ term wrt the weighted input $z$ of the output layer. We can easily derive the derivative of the loss from first principles as shown below:

$$\frac{\partial \mathcal{L}_{\text{cross-ent}}}{\partial m(x_i,\theta)_j} = -\frac{\partial}{\partial m(x_i,\theta)_j}\left(\sum_i^J y_i \log s_i\right) = -\sum_i^J y_i \frac{\partial}{\partial m(x_i,\theta)_j}\log s_i = -\sum_i^J \frac{y_i}{s_i}\frac{\partial s_i}{\partial m(x_i,\theta)_j} \tag{9}$$

$$= -\sum_i^J \frac{y_i}{s_i}s_i \cdot (\mathbf{1}\{i == j\} - s_j), \text{can be easily derived from first principles,} \tag{10}$$

$$= \sum_i^J y_i \cdot s_j - \sum_i^J y_i \cdot (\mathbf{1}\{i == j\}) = s_j \sum_i^J y_i - y_j = s_j - y_j \tag{11}$$

The partial derivative of the cross-entropy loss function wrt output layer parameters has the form:

$$\frac{\partial \mathcal{L}_{\text{cross-ent}}}{\partial m(x_i,\theta)_j} = s_j - y_j \tag{12}$$

For classification tasks, we directly use this analytic form of the derivative and compute it's norm as weights for adaptive and importance sampling.

## B    ALGORITHM DETAILS

---
**Algorithm 3** Subroutine for initialization for Algorithm 1

---
1: **function** INITIALISATION($x$,$y$,$\Omega$,$\theta$,$B$,$q$)  $\quad\leftarrow$ Initialize $q$ in a classical SGD loop
2: $\quad$ **for** $t \leftarrow 1$ **to** $|\Omega|/B$ **do**
3: $\quad\quad$ $x,y \leftarrow \{x_i, y_i\}_{i=(t-1)\cdot B+1}^{t\cdot B+1}$  $\quad\leftarrow$ See all samples in the first epoch
4: $\quad\quad$ $l(x) \leftarrow \mathcal{L}(m(x,\theta),y)$
5: $\quad\quad$ $\nabla l(x) \leftarrow$ Backpropagate($l(x)$)
6: $\quad\quad$ $\langle \nabla L_\theta \rangle(x) \leftarrow \nabla l(x)/B$  $\quad\leftarrow$ Eq. (3)
7: $\quad\quad$ $\theta \leftarrow \theta - \eta \langle \nabla L_\theta \rangle(x)$  $\quad\leftarrow$ Eq. (2)
8: $\quad\quad$ $q(x) \leftarrow$ COMPUTESAMPLEIMPORTANCE($x,y$)  $\leftarrow$ Report per sample importance Algorithm 2
9: $\quad$ **return** $q$,$\theta$

---

## C    ADDITIONAL RESULTS

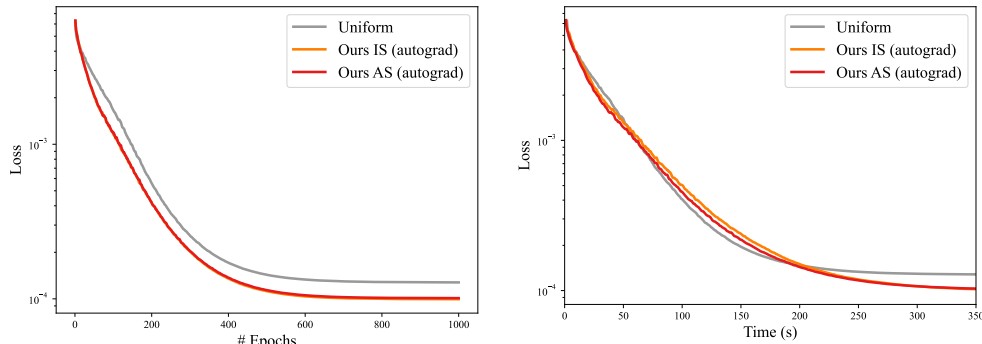

Figure 8: Comparison of the loss evolution for the image regression problem (Fig. 6) at both equal epoch and equal time. While our method have a small overhead cause by the use of autograd, we achieve lower loss for both equal epoch and equal time.

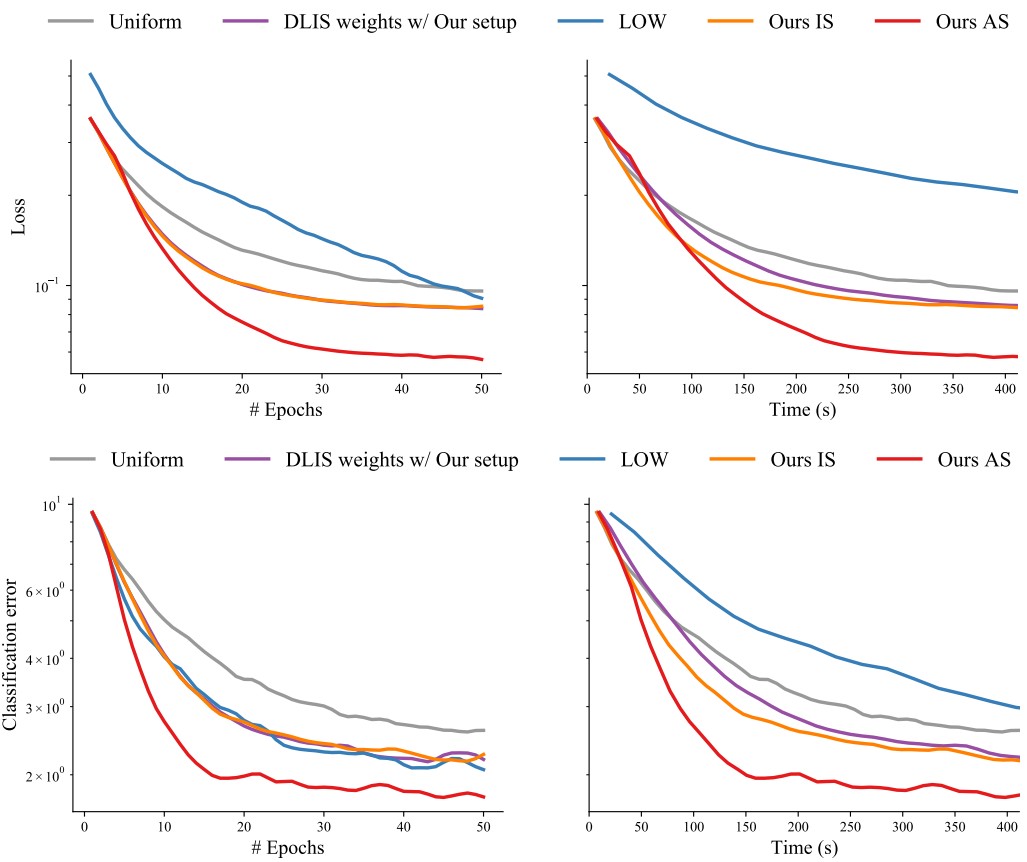

Figure 9: MNIST comparison of adaptive sampling and importance sampling using our method. We compare to DLIS weights using our algorithm and LOW. Loss and Classification error results are presented for equal epoch and equal time. While Ours IS, DLIS, and LOW perform similarly at equal epoch, their computational overhead causes them to perform less at equal time.

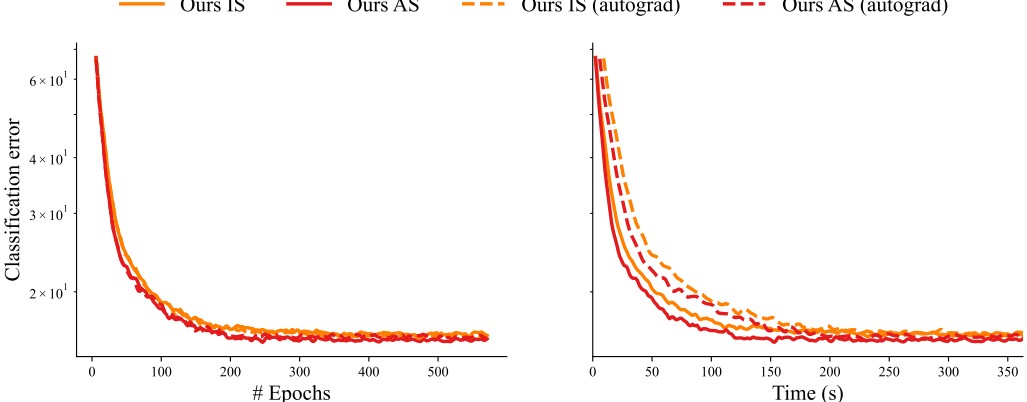

Figure 10: Comparisons between our analytic gradient and autograd numerical gradient on Point cloud classification. Using analytic and autograd gradient results in similar convergence rate and the analytic one is faster without the need of computing per sample gradient using autograd. Meanwhile, it demonstrates that our method is not limited to classification tasks. We show an example in Fig. 6 using autograd gradient for importance and adaptive sampling on a regression problem.

