# OpenReview forum: "Efficient Gradient Estimation via Adaptive and Importance Sampling"
_ICLR.cc/2024/Conference — Submitted to ICLR 2024_

### Official Review · Reviewer_qfPH · 2023-10-21

**Soundness:** 2 fair
**Presentation:** 2 fair
**Contribution:** 2 fair
**Rating:** 3
**Confidence:** 3

**Summary:**

Rather than the usual uniform sampling for stochastic gradient descent, this paper considers adaptive and importance sampling to perform variance reduction by attributing higher sampling probabilities to more important data points. The challenge with implementing adaptive and importance sampling in practice is that computing or even approximating the optimal sampling probabilities for importance sampling essentially requires computing the full gradient, at which point one might as well perform the full gradient descent.

This paper proposes an algorithm that approximates the probabilities for adaptive/importance sampling and then considers a specific approach that uses the cross-entropy loss gradient, e.g., for the purposes of classification tasks.

**Strengths:**

+ Variance reduction in stochastic gradient descent is an important research direction with significant implications and high relevance to the ICLR community.

+ The proposed algorithm improves on uniform sampling in the provided experiments.

+ Experiments conducted on a large number of datasets.

**Weaknesses:**

- Sufficient details have not been provided to formally understand the guarantees of the subroutines in Algorithm 1, such as ComputeImportanceSampling

- No convergence analysis is provided for Algorithm 1 and thus it is difficult to ascertain under what conditions the importance sampling probabilities can be quickly approximated

- It is not clear to me what the general framework in Section 4 is proposing, perhaps additional pseudocode would be helpful

- There are a large number of popular methods for either acceleration or variance reduction for SGD that should be compared for a thorough empirical evaluation (for instance, any number of the references in the related work section)

**Questions:**

- Is there any analysis you can provide for convergence?

- After the approximate probabilities for importance sampling are obtained, how does the algorithm efficiently use these values to sample a data point? That is, are the sampling probabilities organized into a data structure of some sort that maps each real number (possibly implicitly through a set of intervals) to a specific data point?

Update: I acknowledge receipt of the author updates through the discussion phase. I will maintain my score for now but I appreciate the efforts made to improve the quality of the paper.

---

> ### Author Response · Authors · 2023-11-17
>
> # Insufficient details to formally understand the guarantees of Algorithm 1 subroutines; additional pseudocode would be helpful
>
> Please refer to the general response where we detail the algorithm.
>
> The subroutine $\text{ComputeImportanceSampling}$ is intentionally abstract, to represent any importance-sampling heuristic. The purpose of this subroutine is to compute the importance of each mini-batch sample and update the memory vector $q$. We will include pseudo-code implementation for our specific importance function.
>
> # Algorithm 1: no convergence analysis, when importance sampling probabilities can be quickly approximated
>
> Deriving general convergence properties for the algorithm is not feasible as it depends on the sampling distribution. Faster convergence is expected when the distribution closely aligns with the norm of the true gradient. Figure 7 illustrates that our sampling distribution is the closest to the gradient norm.
>
> The computation of sample importance can introduce overhead. Our importance function adds negligible overhead, especially compared to the state of the art. For categorical cross entropy an analytical expression is avilable (Eq.5). Automatic differentiation can be used for other loss functions; its performance impact is also low, as seen in Figure 10.
>
> # Unclear what the framework in Section 4 is proposing
>
> In Section 4 we first analytically derive the importance function for the categorical cross entropy loss. Then we generalize our idea to regression tasks. In particular, we only need to compute the loss gradient w.r.t. the network _output_ to apply our importance sampling strategy to other tasks.
>
> # More SGD acceleration methods to compare to
>
> We have compared our method against state-of-the-art importance and adaptive sampling methods like DLIS, LOW and loss-based importance sampling. Our approach, in principle, can be combined with other variance reduction strategies. However, we believe that this would require additional exploration and goes beyond the scope for this work.
>
> # How does data-point sampling work, based on the the approximate probabilities?
>
> We first normalize the persistent importance distribution $q$ to obtain a probability distribution $p$ (Algorithm 1, L6) that is then used to select data points (L7). Standard routines can be used to perform the importance sampling given $p$. Our implementation uses `numpy.random.choice`; another option is `torch.multinomial`.
>
> Note that when sampling multiple data points simultaneously, it is important to use sampling with replacment, otherwise the prescribed probability distribution $p$ will not be obeyed.

---

### Official Review · Reviewer_FEz4 · 2023-10-29

**Soundness:** 3 good
**Presentation:** 3 good
**Contribution:** 3 good
**Rating:** 6
**Confidence:** 4

**Summary:**

Post rebuttal: I'm raising my score to acceptance. The paper can be strengthened by some theoretical backing, as the paper currently reads that it is a cheaper approximation to Katharopoulous and Fleuret, but doesn't describe _what_ the method actually models.

-----------------------

The paper unifies approaches on importance and adaptive sampling. For this the paper proposes a simple importance function that is logit gradients for which there exists a close form expression for classification networks. They keep track of the importances of the samples across epochs and use that importance to sample the data and compute the weight function $w(x)$.
On multiple varieties of problems like image Classification, image regression, point set classification they show that they converge faster than prior works on importance sampling and adaptive sampling.

**Strengths:**

The paper is well motivated and placed within the literature. The method is intuitive, and the paper is generally written well (see comments below). The biggest strength is the experiments where the authors show wall-clock speedups of their algorithm, instead of some hypothetical quantity. The experiments cover the usual set of benchmarks.

**Weaknesses:**

The only issue I see is with the specific motivation of the proposed importance function. I had a very tough time parsing the text below Eq 5 on page 5. For eg: The gradient norm in eq has two quantities $x$ and $x_i$. Also, the lipschitz constant is written as $l_{m(x, \theta)}$ indicating that the constant of the network is input dependent. Is this intended? Additionally, this bound on the gradients is not informative about the chosen importance function. Eqn 6 makes it seem that the proposed importance function is valid only for Lipschitz neural networks. Is my understanding correct? If so, the proposed method is limited in its applicability.

A discussion on the theoretical properties of the importance function is needed: why is this importance function better than the one in DLIS? In the absence of this, it is tough to attribute the successes of this method. For eg: the way the importances are tracked in Algorithm1 (Lines 13, 14) and the subsequent sampling in Line 7 (instead of the more involved update process in DLIS paper) may be more "implementation friendly" and thus the noticed performance improvements.

Finally, the experiments are limited in the range of experiments covered. The paper covers experiments on MLPs and CNNS. It would interesting to see how the method performs on modern transformers.

**Questions:**

In addition to the comments on the importance function, I have the following questions:
* What is cost of first epoch to estimate a starting importance function? Does that mean that you run a standard SGD loop and track $q$ without any other sampling?
* What is value of $\alpha$ used in Eqn 7? How important is it?
* Why is the DLIS performance on Oxford flowers getting worse after a point? What is sparsity of dataset?
* Last figure on page : there is a reference to "difference between various weighting strategies and the full gradient norm": What is full gradient norm here?

## Minor
* In page 2: paragraph on importance sampling: what is "inherently multivariate" mean?
* Same as above: What is good compromise between optimization quality across all dimensions simultaneously mean? What dimensions?
* The next para, last sentence is incomplete: `Recently, We....`
* Page 3. $N$ is used before it is introduced. Change to $|\Omega|$.
* Last figure on page 9 is uncaptioned.

Overall, I think its a very interesting contribution with good practical use. If the authors can provide a convincing rebuttal, I would be willing to raise my scores.

---

> ### Author Response · Authors · 2023-11-17
>
> # Transformer
>
> We have performed additional experiments on CIFAR-10 using vision transformers (ViT). The results show consistent improvement of ours IS/AS over LOW and DLIS. We will add the plot to the appendix.
>
> # DLIS underperformance on Oxford flower 102
>
> The Oxford flower 102 dataset comprises 102 classes across 1020 images, i.e. 10 images/class, which is sparse compared to other datasets. In this example, DLIS has a poor performce as the mini-batch size is small. This, in combination with a resampling-based algorithm with high class count, yields poor gradient estimation, making it hard for the method to converge.
>
> # Figure 7
>
> In this figure, we refer to the norm of gradient w.r.t. all parameters as the full gradient norm. We use this term to distinguish it from importance distribution based on subsets of network (DLIS, ours).
>
> # Parameter $\alpha$ in Eq.7
>
> We choose $\alpha$ via grid search in $\{0,0.1,0.2,0.3\}$ depending on the task.
>
> # Naming convention
>
> The gradient of a network lives in parameter space. The gradient can be seen as a multivariate vector. We generally refer to the different parameters as dimensions in this parameter space. We will clarify.
>
> The error in the parameter-gradient estimation depends on the sampling strategy. A sampling distribution may yield an estimation that is accurate for parameters and poor for others. A good distribution achieves low error across all parameters at the same time. We refer to this as compromise of quality across dimensions.

---

> > ### Comment · Reviewer_FEz4 · 2023-11-22
> > **Reply to rebuttal**
> >
> > Thank you for the clarifications. Can you also comment _why_ you think your method works better than DLIS which is, arguably, DLIS is better grounded in theory.
> >
> > Also, why does Eq 6 compute the input gradient, and not parameter gradient? The lack of clarity of exposition in this section hinders understanding the method (and not just the pseudocode/algorithm).

---

> ### Author Response · Authors · 2023-11-22
>
> # Imporvement over DLIS
>
> The main distinctions and reasons for improvement lie in the algorithm and the computation of weights. Utilizing the derivative of the loss with respect to the model output is significantly faster than DLIS's importance metric for comparable distributions.
> Additionally, unlike DLIS, our algorithm does not rely on resampling which avoids the computational cost of forward passes on samples without subsequent backward loops.
> These two differences account for the superior performance of our approach at equal time comparison.
>
> Figure 1 shows that both DLIS and our approach have similar importance distribution, but ours is computationally efficient.
>
> Figure 7 further compares the weights. Ours and DLIS weights are closer to the optimal weights.
>
> # Correction of Eqn. 6
>
>
> We regret the typo in this equation. We have corrected this equation and it has the following form:
> $\left\Vert \frac{\partial \mathcal{L}(x_i)}{\partial \theta} \right\Vert = \left\Vert \frac{\partial \mathcal{L}(x_i)}{\partial m(x_i,\theta)} \cdot \frac{\partial m(x_i,\theta)}{\partial \theta} \right\Vert \leq \left\Vert \frac{\partial \mathcal{L}(x_i)}{\partial m(x_i,\theta)} \right\Vert \cdot \left\Vert \frac{\partial m(x_i,\theta)}{\partial \theta} \right\Vert$.
>
> Since the gradient of the loss wrt the parameter $\theta$ is bounded by the gradient wrt the output layer, we chose to importance sample this term: our analytic importance function (shown in eqn. 5).
> The purpose of this equation 6 is to provide a bound on the derivative with respect to the model parameters, not the input. We will add this clarification in the revised version.

---

### Official Review · Reviewer_yYPP · 2023-10-31

**Soundness:** 3 good
**Presentation:** 2 fair
**Contribution:** 3 good
**Rating:** 6
**Confidence:** 3

**Summary:**

The paper proposes a gradient expectation estimation based on adaptive sampling and samples weighting approach. The presented framework is flexible to any function calculating the samples importance. However, the authors propose an efficient importance function based on the loss gradient of the output layer.

**Strengths:**

The presented approach is interesting and is sound.
The overhead required for calculating the weights for the resampling is relatively low making the approach attractive given the shown gain in the overall training time.

**Weaknesses:**

The title of the paper sounds problematic from grammatical point of view as the word adaptive is an adjective unlike importance.
While the sampling scheme is well founded, it seems that the presented approach can suffer from overfitting issues. The authors explicitly propose to add an \epsilon to the value of the importance to avoid focusing on a small set of the data points.

While the presented experiments confirm faster convergence in most cases, it is relatively limited and further discussion on potential drawbacks of the proposed sampling, e.g., robustness to label noise and generalization would have been helpful.

The authors make reference to a paper they presented earlier which should have been avoided given the double blind review process: "Recently, We propose an efficient algorithm and an importance function which when used for importance or adaptive sampling, shows significant improvements."

**Questions:**

What is meant with the sentence: In this method, as outlined in eq. (3), each data point’s weight, denoted as w(xi), remains constant at N. ?

Beyond the weight calculation overhead what are limitations and potential drawbacks of the proposed method?

---

> ### Author Response · Authors · 2023-11-17
>
> # Title choice
>
> We mean "adaptive sampling" and "importance sampling", both of which are standard terms in literature. We are happy fully spell these out in the title, unless reviewers think this would make it overly verbose.
>
> # What is meant with the sentence: In this method, as outlined in eq. (3), each data point’s weight, denoted as w(xi), remains constant at N. ?
>
> In adaptive sampling, achieving a proper average estimator requires the weight $w$ to compensate for the factor $1/|\Omega|$ in front of the summation. Hence, we set the weight to the dataset size: $w(x_i) = |\Omega|$.
>
> # Limitations/drawbacks beyond weight calculation overhead
>
> Our proposed algorithm relies on a memory of the data importance, with size proportional to the dataset size. In cases where data is streamed and not stored, the algorithm cannot be applied. It can also create outdated importance for data if this memory is not updated often enough. This can happen with a high learning rate and result in a poor sampling distribution. In the worse case, the error in the gradient estimation can be higher than without importance sampling.
>
> Another limitation is related to non-uniform data sampling: The samples that go into a mini-batch are not laid out contiguously in memory, which slows down their loading. We believe that a careful implementation can mitigate this issue.
>
> We will add a paragraph at the end of the result section to discuss the limits and drawbacks of the algorithm and sampling strategy.
>
> # Other concerns
>
> From our experiments, the proposed method does not suffer from more overfitting than other methods. In all our presented experiments, we have reported the metric tested on data never seen during training.
>
> Note: The term "Recently" in the reported sentence is a typo and will be corrected.

---

### Official Review · Reviewer_iVwr · 2023-11-06

**Soundness:** 3 good
**Presentation:** 2 fair
**Contribution:** 3 good
**Rating:** 5
**Confidence:** 4

**Summary:**

The authors proposes an sampling strategy that depends on gradient of the loss for training machine learning problems, with both importance and adaptive sampling. It was tested on several classification as well as regression tasks with results that look promising. It demonstrates that by focusing more attention to samples with critical training information, one might be able to speed up convergence without adding computational cost.

**Strengths:**

1. I especially like the visualization of the importance sampling in Figure 1, where 800 data-point are presented with a transparency proportional to their weight according to our method for a classification task. It clearly shows how the algorithm works intuitively.

**Weaknesses:**

The paper can be improved in several ways:

1. The way the authors cite the references sometimes is confusing. It is hard to distinguish the main context from the reference. Please consider use paper numbers only or use parenthesis version.

2. The paper lacks discussions of related paper. For example, https://arxiv.org/pdf/2104.13114.pdf also considers the importance sampling problem by sampling data points proportionally to the loss, instead of norm of gradient.

For another example, https://arxiv.org/pdf/2306.10728.pdf also proposes adaptively sampling methods for dynamically selecting data points for mini-batch. I'd love to see the authors discussed more about these papers.

3. I'd like to see the authors elaborate more on the algorithms. For example, ComputeSampleImportance is mentioned in line 13 without further explained in this section.

4. There seem to be many typos for the math in the paper. For example,

$$\mathcal{L}_{\text {cross-ent }}=-\sum_i y_i \log s_i,$$
 where $s_i=\frac{\exp \left(m\left(x_i, \theta\right)_l\right)}{\sum_l^J \exp \left(m\left(x_i, \theta\right)_l\right)}$. Please correct the subscripts from (8) - (11).

For another example, the explanation after (4) is a bit confusing: where $m\left(x_i, \theta\right)$ is an output layer, $x$ is the input data and $J$ means the number of classes. Try to directly use $x_i$ instead of $x$.

One more example, why does both $x_i$ and $x$ exist in (6)?

$$\left\|\frac{\partial \mathcal{L}(x)}{\partial x}\right\|=\left\|\frac{\partial \mathcal{L}(x)}{\partial m\left(x_i, \theta\right)} \cdot \frac{\partial m\left(x_i, \theta\right)}{\partial x}\right\|$$

5. It is hard to tell the algorithm performance differences for some figures. For example, in Figure 5 (left), the authors claim that at equal epochs (left), our methods (Ours IS & AS) show improvements compared to LOW Santiago et al. (2021) and DLIS weights. It is really invisible to see the difference. The authors may consider plotting the log scale results.

6. There is no reference number to the figure in page 9.

**Questions:**

1. I feel a bit confused in Ours IS vs Ours AS. Does Ours IS mean that you set $w(x) =1 / p(x)$  and Ours AS mean that you set$w(x) =1 / N$? If so, why do you claim your adaptive sampling approach subsumes the adaptive weighting of Santiago et al. in page 3? I think they are different because Ours AS just do the sampling with non-uniform distribution and there is no weighting.

---

> ### Author Response · Authors · 2023-11-17
>
> # Link between adaptive sampling and adaptive weighting (LOW)
>
> Adaptive sampling selects samples non-uniformly while weighting them equally. The same sample can be selected multiple times during training, giving it a higher "weight". Adaptive weighting directly assigns higher weight to more important samples, while selecting the samples uniformly. The two strategies are very similar, and our framework can be tailored to perform either. Applying weighting through sampling is known to achieve better performance than explicit weighting.
>
> # Additional related work
>
> We will discuss the recommended citations in the related work section.
>
> # Figure readability
>
> We will add zoom-ins in Figure 5 to clearly show the differences in convergence behavior near the tails of the plots.

---

> > ### Comment · Reviewer_iVwr · 2023-11-23
> >
> > thank you for the response. Looking forward to seeing the improved version. I will remain the same score at this moment.

---

### Author Response · Authors · 2023-11-17

We thank the reviewers for the insightful feedback. We will carefully consider your remarks. Below we clarify major concerns.

# Algorithm clarification

We begin by clarifying (line-by-line) the details of our proposed Algorithm 1. We will annotate the pseudo-code with comments accordingly.

We propose a general-purpose importance/adaptive sampling algorithm in which existing (or future) importance functions can be plugged in directly.

- L1-3: Parameter initialization. $q$ represents the un-normalized weight (to be) assigned to each sample $x$.

- L4-5: Loops over epochs and mini-batches.

- L6: For each mini-batch $t$, we normalize the weights and store it in an array $p$ that represents the probability density function (PDF).

- L7: $B$ data samples $x_i$ are selected proportionally to the PDF $p$.

- L10: Once the samples are selected (on L7), we assign a weight to each. For importance sampling, the weight is $|\Omega|/p(x_i)$ and for adaptive sampling the weight is constant: $|\Omega|$.

- L11: Monte Carlo estimation of the gradients w.r.t. parameter $\theta$.

- L12: The model parameters $\theta$ are updated via an SGD step.

- L13: $\text{ComputeSampleImportance}(x)$: The un-normalized weights $q(x)$ of the selected samples $x = \{ x_i \}$ are updated according to the sampling strategy. For example, for DLIS sampling strategy, the weights are computed by a metric based on the last model layer. Similarly, for loss-based importance sampling, $\text{ComputeSampleImportance}(x)$ outputs the loss w.r.t. each mini-batch data sample $x_i$. Any existing or future importance sampling functions can be plugged in this line to compute the sample importance, making our algorithm general.

- L14: At the end of each epoch, we add a small value $\epsilon$ to the un-normalized weights $q$ of _all_ data to ensure that every data point will be eventually evaluated, even if its importance is deemed low by the importance metric (on L13). In other words, it ensures no data samples are forgotten indefinitely during the training process.

We will update the description in the paper accordingly.

# Similarities and differences to DLIS

Our importance metric is similar to DLIS (Katharopoulos \& Fleuret 2018) in that it relies on information taken at the end of the network. While this similarity allows us to derive a similar bound for the gradient magnitude, there are two major difference between between the two methods.

Our weights are computed w.r.t. the _output_ of the network instead of its last layer. The resulting importance distributions are similar (as shown in Figures 1 and 7), but our importance function is cheaper to evaluate.

Our proposed algorithm requires storing one scalar importance value per data point, persistently over course of optimization. Unlike resampling algorithms, this avoids computing a forward pass on data without following it up with backpropagation. This results in a more efficient use of the computational resources with small memory overhead.

Figure 7 shows a comparison of various weighting schemes relative to the optimal weights (gradient norm of all parameters); our scheme exhibits best correlation with the optimal weights.

# Mathematical derivation (typos)

We will correct all typos in our equations that where causing confusion. We will also verify the consistency of notation and subscripts.

In Section 4 we show a gradient-norm bound that uses the Lipshitz constant of the network. This derivation is carried out in similar way to DLIS (Katharopoulos \& Fleuret 2018). While it creates a connection to the gradient norm, it does not give indication about the _tightness_ of the bound. This limitation is true for our both method and DLIS. Nethertheless, this connection can still provide insight.

While the bound is derived for Lipshitz networks, our metric can be used for other models.

# Discussion of limitations

We will add a discussion paragraph at the end of the experiments section to elaborate the limitations of our approach.

Our algorithm maintains a record of data importance ($q$). This record is (partially) updated at each mini-batch training step to benefit from the forward loop of the SGD. However, if the previous record is too different from the current one it may lead to an increase in the noise is the gradient estimation.
To avoid the impact of outlier importance values ($q$), we update the importance weights smoothly, using an exponential moving average (Eq. 7).

Thank you,

The Authors

---

### Meta-Review · Area_Chair_vyxd · 2023-12-07

**Metareview:**

The paper presents an importance sampling method for SGD training based on the per-example residuals (loss-gradients wrt the model's output).  In experiments, it is shown that this simple strategy performs comparable or better to recent state-of-the-art methods.

The paper tackles an important problem and is well-motivated. However, the proposed method has limited novelty over previous papers on importance sampling.  While the paper shows encouraging empirical results, if the main focus is on an empirical study then more thorough and larger-scale experiments would be required. Moreover, the presentation of the paper can be improved in many aspects, as pointed out by the reviewers.  The proposed method is very simple, and I suggest to simplify, shorten and streamline the presentation to improve clarity.

Based on the above, my recommendation is to reject the paper at this stage.

**Justification For Why Not Higher Score:**

Since the paper has limited novelty and theoretical results, to recommend acceptance, I would have liked to see an improved and simplified presentation along with larger-scale experiments.

**Justification For Why Not Lower Score:**

N/A

---

### Decision · Program_Chairs · 2024-01-16

Reject